# Traffic and Scenario Adaptive OFDM-IM for Vehicular Networks: A Fuzzy Logic Based Optimization Approach

**DOI:** 10.3390/s25030663

**Published:** 2025-01-23

**Authors:** Xingliang Ren, Yaqi Wei, Lina Zhu, Mohammed Nabil El Korso

**Affiliations:** 1Xidian University, Xi’an 710071, China; rxl@stu.xidian.edu.cn (X.R.); m18734655880@163.com (Y.W.); 2State Key Laboratory of Integrated Services Networks, Xidian University, Xi’an 710071, China; 3CentraleSupélec Laboratoire des Signaux et Systèmes, CNRS, Université Paris-Saclay, 91190 Gif-sur-Yvette, France; mohammed.nabil.el-korso@centralesupelec.fr

**Keywords:** OFDM-IM, vehicular network, adaptive, fuzzy logic

## Abstract

Orthogonal Frequency Division Multiplexing with Index Modulation (OFDM-IM) holds significant importance in vehicle-to-everything (V2X) communications, with its main advantages being outstanding spectral efficiency and strong interference resistance. However, the existing OFDM-IM systems in vehicular networks overlook actual vehicular network channels and the impact of scatterers, thus failing to accurately reflect the system performance. Moreover, these systems focus solely on the bit error rate (BER) and ignore user requirements for low energy consumption and high spectral efficiency. To address these issues, we propose a user demand- and scenario-adaptive OFDM-IM method that optimizes the OFDM-IM index parameter by considering the spectral efficiency, BER, and energy consumption. Firstly, considering non-line-of-sight components and roadside reflectors, we establish a vehicle-to-vehicle (V2V) communication channel model for straight road scenarios. Then, we construct a transmission framework for vehicular network communication using OFDM-IM. Specifically, we develop an energy efficiency maximization formula, in which fuzzy logic is used to adjust the weights of the three performance indicators to meet various environmental and user requirements. In detail, we discuss the minimum signal-to-noise ratio (SNR) required for OFDM-IM to achieve a lower BER than traditional OFDM in various vehicular communication scenarios. Thus, we can make appropriate choices based on the robustness of the simulation results. The simulation results presented in this paper indicate our method’s effectiveness in enhancing the system’s reliability, efficiency, and flexibility.

## 1. Introduction

Orthogonal Frequency Division Multiplexing with Index Modulation (OFDM-IM) is a well-established and mature multi-carrier transmission technique that has been widely recognized for its excellent performance [1,2,3,4]. Compared to traditional OFDM, OFDM-IM offers superior bit error rate (BER) performance due to the incorporation of multi-carrier frequency domain index modulation. By flexibly configuring subcarriers, it can achieve a balance between the BER and spectral efficiency as needed. Additionally, due to its unique structure, OFDM-IM systems exhibit a lower peak-to-average power ratio (PAPR) and reduced sensitivity to frequency offset. Therefore, studying OFDM-IM technology in high-speed vehicular networks is crucial in addressing issues such as signal attenuation and multipath effects in high-mobility scenarios, thereby providing faster and more stable communication services for vehicular networks. Furthermore, OFDM-IM exhibits robust interference resistance, which can mitigate the impact of multipath effects, channel noise, and other interference factors on the communication quality, ensuring efficient and stable communication for vehicles in high-speed motion [5].

In recent years, the application of OFDM-IM in vehicular communications has garnered widespread attention. A subcarrier interleaving scheme, proposed in [6], enhances the performance of traditional OFDM-IM systems by increasing the Euclidean distance between modulation symbols. Furthermore, this interleaved grouping OFDM-IM scheme addresses frequency-selective fading and Doppler effects in vehicle-to-everything (V2X) channels, thereby improving the system’s spectral efficiency and reliability, as discussed in [4]. This method was further expanded to independently execute index modulation in the in-phase and quadrature domains, enhancing the BER performance and making the improved OFDM-IM scheme more suitable for mobile scenarios in vehicular networks, according to [7]. The integration of inter-carrier interference (ICI) self-cancellation technology into the OFDM-IM framework, which addresses the issue of subcarrier orthogonality being easily disrupted by Doppler-induced ICI in V2X channels, was presented in [8]. An IM-based multiple-input multiple-output (MIMO) OFDM system was proposed in [5]. Due to the incorporation of IM, this system achieved ICI mitigation and spatial diversity utilization in vehicle-to-vehicle (V2V) channels, while also reducing the complexity and improving the BER performance compared to traditional MIMO-OFDM systems. Compared to traditional OFDM-IM systems, the IM-based MIMO-OFDM system utilizes spatial diversity through MIMO technology, extending index modulation to the spatial domain. This allows for enhanced spectral efficiency and improved robustness against multipath interference, making it better suited for complex vehicular network scenarios. However, the IM-based MIMO-OFDM system also introduces additional complexity in signal processing and channel estimation due to the integration of MIMO, which may limit its practicality in scenarios with stringent real-time constraints or limited computational resources.

Despite the significant progress in enhancing systems’ transmission performance, reducing systems’ complexity, and improving the interference resistance, the application of OFDM-IM systems in practical vehicular network scenarios still faces numerous challenges. Particularly in complex vehicular network environments, accurately modeling OFDM-IM systems and flexibly configuring subcarriers based on specific scenarios and user requirements to optimize system performance remain critical issues. Additionally, with the increasing severity of the energy crisis and the rising market share of new energy vehicles, the study of efficient and reliable communication technologies in vehicular network scenarios must consider the diverse needs of electric and hybrid vehicles regarding low energy consumption and high spectral efficiency [9]. Existing OFDM-IM systems primarily use a low bit error rate as the evaluation criterion, neglecting other key performance indicators, such as the spectral efficiency and energy consumption. Aiming to explore the application of OFDM-IM technology in vehicular network scenarios, we establish an OFDM-IM transmission model that aligns with the characteristics of vehicular networks in this paper. We propose a user demand- and scenario-adaptive OFDM-IM method that considers not only the BER but also comprehensively accounts for the system’s spectral efficiency and energy consumption. By employing fuzzy logic, the optimal OFDM-IM index parameter combinations for different vehicular network scenarios are derived, providing an effective technical solution for the achievement of novel, efficient communication systems in vehicular networks.

The rest of this paper is organized as follows. Section 2 presents the related work. Section 3 establishes a V2V communication transmission model based on OFDM-IM technology in vehicular networks. Section 4 proposes a user demand- and scenario-adaptive OFDM-IM method for vehicular network scenarios. Section 5 verifies the effectiveness of the proposed method through simulation experiments. Finally, Section 6 summarizes the entire paper.

## 2. Related Works

The concept of OFDM-IM was first introduced in [10], where subcarrier index modulation was incorporated into traditional OFDM. The receiver structure for OFDM-IM systems was improved in [1], and a more efficient detection algorithm was proposed. Subsequently, multidimensional index modulation, which utilizes both the spatial and frequency domains for index modulation, was extensively studied, significantly enhancing the transmission capacity and interference resistance, as highlighted in [6]. A distributed processing scheme for multi-relay-assisted OFDM-IM was proposed in [11], providing optimal error performance for OFDM-IM systems. The combination of Multiple-Mode Orthogonal Frequency Division Multiplexing with Index Modulation (MM-OFDM-IM) and MIMO was discussed in [12], where a novel multiple-input multiple-output MM-OFDM-IM (MIMO-MM-OFDM-IM) scheme was proposed and a closed-form theoretical analysis of the BER was derived to evaluate the performance of MIMO-MM-OFDM-IM. The research in [13] investigated the BER performance of OFDM-IM under different mainstream 5G modulation schemes and various detection methods. In the research in [14], an optimized greedy detection scheme was proposed. These studies have achieved significant theoretical as well as technical advancements and provide a solid foundation for the development of OFDM-IM technology.

In vehicular networks, utilizing wireless communication technology to optimize system performance is crucial [15,16]. Additionally, the foundational study by [17] provides critical insights into the parameter configurations and performance characteristics of conventional OFDM systems over V2V channels. Incorporating these findings strengthens the basis for the evaluation and comparison of the proposed OFDM-IM system. This requires the provision of efficient, stable, and secure communication services for numerous vehicles in complex environments to support autonomous driving, V2X communication, and traffic management. The current OFDM-IM systems primarily focus on achieving a low BER as the evaluation criterion, often neglecting other important factors, such as the spectral efficiency and power consumption, in practical implementations [18,19,20,21]. While multidimensional index modulation, which utilizes both the spatial and frequency domains for index modulation, has been extensively studied and shown to significantly enhance systems’ transmission capacity in theoretical research, these advancements are not always fully realized in existing systems. Many of these studies emphasize BER improvements but fail to address the practical trade-offs required to optimize the energy efficiency and spectral efficiency in real-world scenarios. Therefore, a comprehensive performance evaluation should include the spectral efficiency, BER, and power consumption to maximize the system’s effectiveness. Vehicle energy efficiency is a key focus in automotive engineering, aiming to reduce the energy consumption for sustainable development [22,23]. Optimization strategies based on fuzzy logic have significant advantages in improving vehicles’ energy efficiency [24,25]. In recent years, research on electric vehicles (EVs) and energy efficiency optimization has garnered great attention, with companies and research institutions actively exploring fuzzy logic-based energy efficiency optimization technologies. These technologies are mainly applied to predict the future energy consumption of vehicles, optimize systems’ ranges and safety, and achieve optimal energy consumption under various conditions through fuzzy logic control. The authors of [26] proposed a distributed energy-aware fuzzy logic routing algorithm, capturing the network status using energy metrics and mapping them to appropriate cost values, as well as calculating the shortest path based on the predicted energy demands to address energy efficiency and balance issues. In [27], the researchers introduced an energy maximization and management method for hybrid systems, optimizing generator usage through fuzzy logic and developing control and supervision strategies for hybrid power systems. Overall, with the increasing prevalence of electric vehicles and growing attention given to energy efficiency, efficient and reliable communication technologies in vehicular networks must also meet the energy demands of the EV market to achieve sustainable development and maximize the energy benefits.

Therefore, based on the basic principles of OFDM-IM technology, this paper models the OFDM-IM system in vehicular network scenarios and proposes a user demand- and scenario-adaptive OFDM-IM method. This method employs fuzzy logic to adjust the allocation of performance metrics according to user needs in different scenarios, thereby obtaining the optimal parameter configuration for the OFDM-IM system.

## 3. System Model

In this paper, we primarily focus on vehicular network scenarios and select the most representative straight road scenario for modeling [28,29,30,31]. In this scenario, vehicles travel in opposite directions along a straight road, utilizing OFDM-IM technology to achieve V2V communication. We define the communication model and OFDM-IM transmission framework for this scenario. By considering the impact of scatterers on both sides of the road on the signal propagation path and intensity, we analyze in detail the contributions of line-of-sight (LOS) and non-line-of-sight (NLOS) path components to channel fading. Additionally, we provide the transmission framework and transceiver module model for this system, laying the foundation for subsequent simulations.

### 3.1. V2V Scenario Model for Straight Roads

As shown in Figure 1, buildings and trees are the main scatterers distributed along the sides of the road. The horizontal distance between the signal-transmitting vehicle ($T_{X}$) and the receiving vehicle ($R_{X}$) is D, and the straight-line distance is $D_{0}$. The distance between the scatterer S and the transmitter is hr1, and the distance between the $R_{X}$ and the $T_{X}$ is hr2. Both the transmitting and receiving vehicles use a uniform single antenna. The angle of departure (AOD) of the signal is $\alpha_{T_{1}}$, the angle of arrival (AOA) is $\alpha_{R_{1}}$, the angle of the direct component at the transmitter is $\alpha_{T_{0}}$, and the angle of the direct component at the receiver is $\alpha_{R_{0}}$. The speed of the transmitting vehicle is $V_{T_{x}}$, and the speed of the receiving vehicle is $V_{R_{x}}$, with their respective directions of travel being $\beta_{T}$ and $\beta_{R}$. The scatterers on both sides of the road affect the strength and propagation path of the signal received by the receiving vehicle. The received signal includes not only the LOS component but also a large number of NLOS components that arrive after being reflected by the scatterers. This paper considers only single scatterer reflections. The parameters of the channel model are summarized in Table 1.

### 3.2. V2V Channel Model for Straight Road Scenario

In the channel transmission model, considering the frequency non-selective fading channel, the complex channel fading envelope between the transmitting antenna and the receiving antenna can be expressed as [31](1)$$h_{TR}\left(t\right)=h_{TR}^{LOS}\left(t\right)+h_{TR}^{NLOS}\left(t\right).$$

#### 3.2.1. Modeling of the LOS Component

According to Figure 1, $D_{TR}^{LOS}$ represents the LOS propagation path between $T_{X}$ and $R_{X}$. Thus, the channel fading of the direct path $h_{TR}^{LOS}$ can be expressed as follows: (2)$$h_{TR}^{LOS}\left(t\right)=\sqrt{R/(R+1})*e^{-j2\pi f_{c}D_{TR}^{LOS}/c}*e^{j2\pi t(f_{T}^{LOS}+f_{R}^{LOS})},$$
where *R* is the Rician factor, representing the ratio of the average power of the LOS component to the total power of all NLOS components, used to characterize the strength of multipath components during signal transmission. *c* is the speed of the wave, which is approximately $3\cdot 10^{8}$ m/s. $f_{c}$ is the carrier frequency, while $f_{T}^{LOS}$ and $f_{R}^{LOS}$ are the Doppler shifts caused by the movement of the transmitting and receiving vehicles, respectively, affecting the LOS components. The specific formulas are given as follows: (3)$$f_{R}^{LOS}=f_{c}v_{R_{x}}\cos\left(\alpha_{R_{0}}-\beta_{R}\right)/c,$$(4)$$f_{T}^{LOS}=f_{c}v_{T_{x}}\cos\left(\alpha_{T_{0}}-\beta_{T}\right)/c.$$

#### 3.2.2. Modeling of NLOS Components

If we consider a type of scatterer at the roadside and only account for single reflections, $D_{TS}^{NLOS}-D_{SR}^{NLOS}$ forms a complete NLOS path $D_{TR}^{NLOS}$, and the channel fading $h_{TR}^{NLOS}$ caused by the NLOS path can be expressed as follows: (5)$$h_{TR}^{NLOS}\left(t\right)=\sqrt{\frac{1}{R+1}}\mu \cdot e^{j(\theta -2\pi f_{c}D_{TR}^{NLOS}/c)}\cdot e^{j2\pi t(f_{T}^{NLOS}+f_{R}^{NLOS})},$$(6)$$f_{R}^{NLOS}=f_{c}v_{R_{x}}\cos\left(\alpha_{R_{1}}-\beta_{R}\right)/c,$$(7)$$f_{T}^{NLOS}=f_{c}v_{T_{x}}\cos\left(\alpha_{T_{1}}-\beta_{T}\right)/c,$$
where μ is the reflection coefficient of the scatterer *S*’s surface. In various vehicular network scenarios, roadside scatterers have different material compositions and quantities, leading to different reflection coefficients. θ is the random phase generated by the electromagnetic wave on the scatterer, following a uniform distribution within [0, 2π].

#### 3.2.3. Scenario Assumptions

Assume that $T_{X}$ and $R_{X}$ are moving towards each other in two lanes at a speed of 14 m/s. Considering only one type of scatterer, the most common trees are selected as scatterers. The remaining relevant parameters are set as follows: $D_{0}$ = 100 m, D = 71.4 m; hr1 = 20.8 m, hr2 = 37 m; $\alpha_{T_{0}}$ = −21.9°, $\alpha_{R_{0}}$ = 158.73°, $\alpha_{T_{1}}$ = 15°, $\alpha_{R_{1}}$ = 105.63°; $f_{c}$ = 6 GHz. We substitute these values into the formulas provided in the previous two sections to calculate(8)$$h_{TR}^{LOS}\left(t\right)=\sqrt{R/(R+1})e^{-j4000\pi }e^{j1041.6\pi t},$$(9)$$h_{TR}^{NLOS}\left(t\right)=\sqrt{1/(R+1)}\mu e^{j(\theta -5600\pi )}e^{j692.16\pi t}.$$

By substituting (Equation 8) and (Equation 9) into (Equation 1), the final expression for the complex channel fading envelope is obtained as(10)$$h_{TR}\left(t\right)=\sqrt{\frac{R}{R+1}}e^{-j4000\pi }e^{j1041.6\pi t}\;+\sqrt{\frac{1}{R+1}}\mu e^{j(\theta -5600\pi )}e^{j692.16\pi t}.$$

At this point, the only unknowns in expression $h_{TR}\left(t\right)$ are the scatterer reflection coefficient μ and the Rician factor R. In different vehicular network scenarios, μ and R are key parameters affecting signal propagation, and their variations directly determine the characteristics of channel fading in each scenario. By substituting the values of μ and R in different scenarios into (Equation 10), we can construct a more accurate system model.

The time-domain channel gain $h_{TR}\left(t\right)$ described by (Equation 1)–(Equation 10) represents a simplified scenario in vehicular communication, where the channel gain is considered to vary only with time *t*, without explicitly capturing the multipath delay characteristics. This model is suitable for scenarios where the bandwidth is relatively narrow. However, in practical wideband multipath environments, frequency selectivity plays a crucial role in determining the performance of OFDM systems. To bridge the gap to the OFDM-IM system model presented later, we need to extend $h_{TR}\left(t\right)$ to a channel model that includes a time-varying impulse response with multipath delays:(11)$$h_{TR}\left(\tau ,t\right)=\sum_{p=1}^{P}\alpha_{p}\left(t\right)e^{j\phi_{p}\left(t\right)}\delta \left(\tau -\tau_{p}\right),$$
where $\alpha_{p}\left(t\right)$ and $\phi_{p}\left(t\right)$ represent the time-varying amplitude and phase of the *p*-th path, $\tau_{p}$ is the path delay, and *P* is the number of paths. This formulation models the channel as the superposition of multiple delayed components, leading to frequency-selective fading.

If multipath effects are negligible or simplified to a single path, and the delay of this path can be considered zero (or negligibly small), $h_{TR}\left(\tau ,t\right)$ reduces to a function without explicit dependence on τ. Thus, the original $h_{TR}\left(t\right)$ is a special case of $h_{TR}\left(\tau ,t\right)$ under certain conditions. If *P* = 1 and $\tau_{1}\approx 0$, then(12)$$h_{TR}\left(t\right)\approx \int h_{TR}\left(\tau ,t\right)d\tau =\alpha_{1}\left(t\right)e^{j\phi_{1}\left(t\right)}.$$

When considering the multipath structure shown in (Equation 11), the channel is no longer a narrowband gain function solely in time; instead, it becomes a two-dimensional function of the delay and time. When an OFDM system is introduced, along with a cyclic prefix and DFT operations, the multipath convolution in the time domain can be transformed into independent subcarrier gains in the frequency domain.

### 3.3. OFDM-IM Communication Transmission Framework

The block diagram of the OFDM-IM transceiver module model for vehicular communication is illustrated in Figure 2. As shown in this figure, Module A denotes the bit demultiplexer, Module B denotes index selection, Module C denotes constellation mapping, Module D denotes the OFDM block generator, Module E denotes the N-point inverse fast Fourier transform (IFFT), Module F denotes cyclic prefix (CP) addition, Module G denotes CP removal and analog-to-digital conversion, Module H denotes the N-point FFT, Module I denotes signal decomposition, Module J denotes index demodulation, Module K denotes inverse index mapping, Module L denotes M-order demodulation, and Module M denotes parallel-to-serial conversion.

At the transmitter, each OFDM symbol employs *U* subcarriers to transmit a total of *B* bits. These *B* bits are then divided into *G* groups, each containing *p* bits, which gives B=pG. For each group, *K* bits are mapped to an OFDM subblock of length *N*, where N=U/G, and *U* also denotes the size of the fast Fourier transform (FFT). Unlike traditional OFDM, in each subblock, only *K* out of the *N* subcarriers are activated and used for constellation symbol transmission. These *K* subcarriers are termed active subcarriers, while the remaining (N−K) subcarriers are inactive. The *p* bits within each subblock are divided into two parts: $p_{1}$ bits are used to control the positions of the active subcarriers, while $p_{2}$ bits are used for traditional signal constellation mapping.

For each subblock, the input $p_{1}$ bits are transmitted to the index selector. The index selector selects *K* active indices from *N* available indices based on a predetermined index selection scheme. The selected indices can be expressed as(13)$$I_{\beta }=\left\{i_{\beta ,1},i_{\beta ,2},i_{\beta ,3},\cdots ,i_{\beta ,K}\right\}.$$

Among them, $i_{\beta ,k}\in \left[1,2,3,\cdots ,N\right]$, β=1,2,3,⋯,G and k=1,2,3,⋯,K. The number of bits carried by the positions of active indices in the subblock is $p_{1}=\left\lfloor \log_{2}C_{N}^{K}\right\rfloor$. The input $p_{2}$ bits are transmitted to an *M*-bit modulator and mapped to the *M*-bit signal constellation, determining the modulation symbols carried by the *K* active subcarriers. The output of *M*-order constellation modulation is given by(14)$$S_{\beta }=\left[s_{\beta ,1},s_{\beta ,2},\cdots ,s_{\beta ,k},\cdots ,s_{\beta ,K}\right],$$
where $s_{\beta ,k}\in \Omega$ and Ω represent the set of *M*-order constellation signals. Assume that the signal constellation is normalized to unit average power, meaning $E\left\{S_{\beta }S_{\beta }^{H}\right\}=K$. The number of information bits carried by the *M*-order constellation symbols is $p_{2}=K\log_{2}M$. The output of the βth OFDM-IM subblock generator is given by(15)$$X_{\beta }={\left[x_{\beta }\left(1\right),x_{\beta }\left(2\right),\cdots ,x_{\beta }\left(n\right),\cdots x_{\beta }\left(N\right)\right]}^{T},$$(16)$$x_{\beta }\left(n\right)=\left\{\begin{array}{c} s_{\beta ,n},n\in I_{\beta }\\ 0,o\mathrm{therwise} \end{array}\right.,$$
where n=1,2,⋯,N. The OFDM-IM subblock generator takes into account $I_{\beta }$ and $S_{\beta }$ for each subblock. After generating all OFDM-IM subblocks, these subblocks are combined into an U*1-dimensional OFDM-IM block.(17)$$X_{F}={\left[x(1),x(2),\cdots ,x(u),\cdots ,x(U)\right]}^{T}.$$

To counteract channel fading, $x\left(u\right)\in \left\{0,\Omega \right\},u=1,2,\ldots ,U$ can apply block interleaving to $X_{F}$, resulting in an interleaved OFDM-IM block, ${\tilde{X}}_{F}$. Then, ${\tilde{X}}_{F}$ can be converted to the time domain using an IFFT.(18)$${\tilde{X}}_{T}=\frac{U}{\sqrt{KG}}\mathrm{IFFT}\left\{{\tilde{X}}_{F}\right\}=\frac{1}{\sqrt{KG}}W_{U}^{H}{\tilde{X}}_{F}.$$

${\tilde{X}}_{T}$ is the time-domain OFDM-IM block, $W_{U}$ is the FFT matrix and $w_{U}^{H}W_{U}=UI_{U}$, $U/\sqrt{GK}$ is the normalization factor for the FFT, and $E\left\{{\tilde{X}}_{T}^{H}{\tilde{X}}_{T}\right\}=U$. At the transmitter, a cyclic prefix (CP) with a length greater than the maximum channel delay must be added to ${\tilde{X}}_{T}$ before transmission.

At the receiver, after the signal passes through the wireless fading channel, it undergoes CP removal, FFT, and deinterleaving; the frequency domain data can be represented as(19)$$y_{F}=HX_{F}+n_{F},$$
where $y_{F}={\left[y_{F}\left(1\right),y_{F}\left(2\right),\cdots ,y_{F}\left(u\right),\cdots y_{F}\left(U\right)\right]}^{T}$, H is a *U*-dimensional diagonal matrix with its diagonal elements representing the channel impulse response (CIR) coefficients of the multipath channel in the frequency domain, and $n_{F}$ denotes Gaussian noise in the frequency domain with a mean of 0 and variance of $\sigma_{F}^{2}$.

Based on the estimated CIR, channel equalization is applied to $y_{F}$, which can be represented as ${\hat{y}}_{F}$. To detect the positions of the active subcarriers in each subblock, the OFDM-IM signal separator divides ${\hat{y}}_{F}$ into *G* subblocks, with the βth subblock represented as(20)$${\hat{y}}_{\beta }={\left[{\hat{y}}_{\beta }\left(1\right),{\hat{y}}_{\beta }\left(2\right),\ldots ,{\hat{y}}_{\beta }\left(n\right),\ldots ,{\hat{y}}_{\beta }\left(N\right)\right]}^{T},$$
where ${\hat{y}}_{F}$ is passed to the log-likelihood ratio (LLR) detector to determine the positions of the active subcarriers. Subsequently, index inverse mapping and constellation symbol demodulation are performed, generating index information with a length of $p_{1}$ bits and constellation information with a length of $p_{2}$ bits. Finally, the bit combiner merges these into an information bit sequence of length *B*.

To connect the time-domain channel characteristics described by (Equation 1)–(Equation 11) with the frequency-domain representation in (Equation 20), we consider that, within the duration of one OFDM symbol $T_{\mathrm{sym}\;}$, the channel varies slowly. Hence, we assume that $h_{TR}\left(\tau ,t\right)$ is approximately time-invariant during one OFDM symbol, i.e., $h_{TR}\left(\tau \right)\approx h_{TR}\left(\tau ,t\right)$ for $t\in \left[t_{0},t_{0}+T_{\mathrm{sym}\;}\right]$. Sampling this time-invariant impulse response in the discrete-time domain (corresponding to *U* points for IFFT/FFT) yields a sequence $h_{TR}\left[l\right]$, l=0,1,…,L−1. The DFT of the discrete-time impulse response $h_{TR}\left[l\right]$, yielding the frequency-domain gains $H_{k}$ for each subcarrier, is shown as(21)$$H_{k}=\sum_{l=0}^{L-1}h_{TR}\left[l\right]e^{-j\frac{2\pi kl}{U}},k=0,1,\ldots ,U-1.$$

Arranging $H_{k}$ into the diagonal matrix $H=\mathrm{diag}\left(H_{0},H_{1},\ldots ,H_{U-1}\right)$ corresponds directly to the frequency-domain model $y_{F}=HX_{F}+n_{F}$ in (Equation 20).

## 4. Adaptive OFDM-IM Method

The proliferation of vehicular network scenarios, coupled with the increasing number of users, intensifies network competition, resulting in scarce spectrum resources. As the number of users continues to rise, the energy consumption due to user mobility in various scenarios must also be taken into account. Building on the previous section, this section further explores the multidimensional requirements for the optimization of vehicular communication systems’ performance. Unlike traditional approaches with a single optimization objective, optimizing vehicular network performance requires attention to be paid not only to the error rate but also to the spectral efficiency and energy consumption, ensuring efficient operation under various interference conditions. Therefore, this section integrates three performance indicators: the spectral efficiency, error rate, and energy consumption. We propose a method to adaptively control the OFDM-IM index parameters according to the scenario and user demands, aiming to maximize the efficiency. By referencing fuzzy logic to calculate the weight distribution, we determine the optimal OFDM-IM index parameters in various scenarios, enabling the adaptive control of the index parameters.

Figure 3 illustrates the overall process of the user demand- and scenario-adaptive OFDM-IM method. First, the channel parameters for the vehicular network scenario are determined, including the reflection factor of NLOS paths and the Rician factor. These parameters are then used to construct the channel formulas for the straight road scenario discussed in the previous section, which are incorporated into the OFDM-IM model to facilitate communication in the vehicular network channel. Subsequently, an energy efficiency maximization formula is established for this scenario, and the user requirements are analyzed to determine the weights of the three factors accordingly. The BER performance under the OFDM-IM model is analyzed to find the minimum signal-to-noise ratio (SNR) at which the OFDM-IM BER surpasses that of traditional OFDM. Using the concept of fuzzy logic, the optimal index parameters of the OFDM-IM model in this scenario are determined to maximize the system’s energy efficiency. The optimal number of subcarriers N, the number of active subcarriers K, and the modulation order M at the minimum SNR for this scenario are identified, thereby implementing a method for adaptive control according to the user requirements and the scenario.

### 4.1. Scenario Model

In practical applications, vehicular network technology needs to adapt to different environments, such as urban traffic scenarios and dense building areas, to meet the user requirements in various mobile contexts. Furthermore, as users move through these scenarios, variations in Doppler shifts, the proportion of direct data components, and the reflection coefficients of roadside reflectors impact the channel quality.

#### 4.1.1. OFDM-IM Communication in Urban Traffic Scenarios

In urban traffic scenarios, as illustrated in Figure 4, in addition to the LOS path, the NLOS components are reflected by scatterers such as trees, with the reflection coefficient of the trees being μ=0.5. At this point, the Rician factor of the channel model is 10 dB. Considering the good channel quality, convenience of charging electric vehicles, and users’ requirements for communication reliability, high network throughput, and low energy consumption in urban scenarios, this study comprehensively examines the performance metrics of the BER, spectral efficiency, and energy consumption. It aims to identify the optimal parameter balance to enhance the system’s performance and meet diverse demands.

#### 4.1.2. OFDM-IM Communication in Highway Scenarios

In highway scenarios, as illustrated in Figure 5, the received signal is primarily composed of LOS components, while the NLOS components are relatively minimal, with a Rician factor of 20 dB. The reflection coefficients of the scatterers along both sides of the road are relatively low, denoted as μ=0.2. In this scenario, the good channel quality contrasts the inconvenience of charging electric vehicles, raising concerns about energy consumption. Therefore, the optimization strategy focuses on minimizing the energy consumption while considering the BER and spectral efficiency, selecting parameters that provide the best overall performance to meet the specific requirements of the OFDM-IM system in highway scenarios.

In vehicular networks, optimizing the performance of the OFDM-IM communication system to meet the requirements of various users and scenarios necessitates consideration of three key factors: the BER, spectral efficiency, and energy consumption. For instance, in highway scenarios, roadside reflectors are sparse and the roads are open, but charging is more challenging than on regular roads. Considering users’ difficulties with charging, the focus should be on optimizing the energy consumption and BER, accepting some loss in spectral efficiency to enhance the overall system performance.

### 4.2. Establishment of the Energy Efficiency Maximization Equation

Due to the dynamic and complex nature of vehicular network environments, traditional fixed-weight evaluation methods cannot accurately assess systems’ energy efficiency. Furthermore, the interdependence among performance metrics makes finding the optimal configuration more challenging. To address these issues, we use the spectral efficiency, BER, and energy consumption as evaluation criteria for the system’s energy efficiency. By establishing the energy efficiency maximization equation, as shown in (Equation 22), it is possible to optimize the multi-carrier index modulation system model for vehicular network scenarios based on the practical setting and user requirements.(22)$$\max Y\left\{H_{i}\left(R_{i},\mu_{i}\right)\right\}=a_{i}\eta +b_{i}BER+c_{i}NH,$$
where *i* represents different scenarios (i={1,2}):i=1 refers to the urban traffic scenario with scenario parameters R=10dB,μ=0.5; i=2 refers to the highway scenario with scenario parameters R=20dB,μ=0.2. Different values of *i* define different channels based on the scenario parameters. *Y* is the output value of fuzzy inference under different typical channel conditions and is used to represent the system’s energy efficiency. $a_{i}$, $b_{i}$, and $c_{i}$ are the logical output weights of three logical factors under different channel scenarios, and η is the ratio of the spectral efficiency of the OFDM-IM system to that of the OFDM system, which is used for comparison with traditional OFDM systems. BER is the system’s bit error rate, and NH is the system’s energy consumption.

It is important to note that the metrics η, BER, and NH in (Equation 22) have been normalized during the fuzzification step using membership functions, ensuring dimensional consistency and enabling a fair comparison of the weighted contributions of each metric.

### 4.3. Determination of Minimum SNR

In vehicle-to-vehicle communication scenarios within the Internet of Vehicles (IoV), the complexity and dynamic nature of the environment usually restrict the range of the SNR, significantly impacting the design and performance of communication systems. Particularly in urban traffic or congested highway environments, the SNR may be limited due to multipath fading, reflections, and other interference. Thus, to optimize the communication performance in the IoV, particularly under these limited SNR conditions, it is crucial to determine the minimum SNR at which the BER of OFDM-IM can surpass that of traditional OFDM technology. This analysis will not only provide insights into the potential of OFDM-IM in the IoV but also offer valuable guidelines for the design and parameter optimization of OFDM-IM methods based on user needs and scenario adaptation.

Before determining the minimum SNR, it is crucial to observe the trend of the BER as the SNR changes by controlling other variables. In Figure 6, we fix the number of subcarriers per subblock N = 8 and analyze the BER performance of OFDM-IM under different modulation orders M and numbers of active subcarriers K. As M increases from 2 to 16, the minimum SNR required for the BER of OFDM-IM to surpass that of traditional OFDM gradually increases. Specifically, the intersection point shifts from approximately 12 dB in Figure 6a to around 20 dB in Figure 6d. Within each subplot, when K = 8 (the number of active subcarriers equals the number of subcarriers in a subblock), the curve represents traditional OFDM. All other curves correspond to OFDM-IM. This comparison clearly demonstrates that, as M increases, achieving better BER performance with OFDM-IM relative to traditional OFDM requires higher SNR levels.

Next, we fix the modulation order M at 4 and vary the number of subcarriers N and the number of active subcarriers K in each subblock to conduct further analysis. As illustrated in Figure 7, with M remaining constant, the minimum SNR required for OFDM-IM to surpass that of traditional OFDM increases as N increases. Based on these findings, we select N = 16 and M = 16 to pinpoint the SNR more precisely. This parameter configuration will encompass all possible combinations of (N, K, M) in this scenario.

In the urban traffic scenario, where the reflection coefficient μ=0.5 and the Rice factor R is 10 dB, the trend of the BER as a function of the SNR is as illustrated in Figure 8, with N = 16 and M = 16. The minimum SNR required for the BER of OFDM-IM to surpass that of traditional OFDM is 16 dB. To enhance the robustness of the simulation, an SNR of 20 dB is chosen.

In the highway scenario, with the reflection coefficient μ=0.2 and the Rice factor R at 20 dB, the trend of the BER as a function of the SNR is as illustrated in Figure 9, with N = 16 and M = 16. Similarly, the minimum SNR required for the BER of OFDM-IM to surpass that of traditional OFDM is determined to be 15 dB. 

To further clarify the results, it should be emphasized that the selection of 20 dB as a reference SNR level in this study does not imply that such a level can be easily attained in practical vehicular communication environments. Instead, this value is chosen as a theoretical benchmark to illustrate the upper bound of the system performance under ideal channel conditions. In real-world scenarios, achieving a stable 20 dB SNR is challenging due to factors such as multipath fading, reflections, interference, and high mobility. Nevertheless, modern communication techniques such as intelligent beamforming, multi-antenna cooperation, and adaptive modulation and coding are gradually enabling the temporary realization of higher SNR levels under controlled conditions or specific scenarios. Consequently, including a 20 dB SNR scenario in our analysis helps to delineate the theoretical performance ceiling of the proposed system and provides a valuable reference point for future research and technological refinements.

### 4.4. Fuzzy Logic Optimization

In this section, we employ the concept of fuzzy logic to optimize the performance of the OFDM-IM system. Fuzzy logic expresses information through linguistic variables rather than numerical ones, facilitating the handling of systems with inherent ambiguities in their descriptions and decision-making processes [32,33,34]. Based on this, we propose a method to adaptively adjust the key parameters of OFDM-IM, such as the number of subcarriers, the number of active subcarriers, and the modulation order, according to different scenarios and user requirements. The core of this method lies in appropriately weighting the three performance indicators—the spectral efficiency, bit error rate, and energy consumption—based on scenario-specific requirements to maximize the overall performance.

Fuzzy logic is a reasoning method based on fuzzy set theory, aiming to simulate human reasoning by processing fuzzy information. Its core framework consists of three main steps: fuzzification, fuzzy inference, and defuzzification. The fuzzification process converts the precise values of input variables into membership degrees, mapping the variables to corresponding fuzzy sets using membership functions. Membership functions are typically represented in forms such as triangular, trapezoidal, or Gaussian curves to quantify the fuzziness of the input values. For example, for the fuzzy set “low BER”, a specific triangular membership function can be defined to map the BER values to membership degrees ranging from 0 to 1. During the fuzzy inference phase, the system performs logical deduction on the input variables based on predefined fuzzy rules (commonly in the form of IF/THEN rules). For instance, the rule “IF energy consumption is low AND BER is low THEN performance is excellent” combines membership degrees to derive the fuzzy set of output variables. Finally, defuzzification converts the output fuzzy set into a specific numerical value, with commonly used methods including the maximum membership method, weighted average method, or centroid method, providing clear decision support for system optimization. By designing appropriate membership functions and fuzzy rules, fuzzy logic effectively handles complex nonlinear relationships among multidimensional input variables, making it particularly suitable for optimizing the OFDM-IM communication parameters in this study. The selection of membership functions and fuzzy rules should be adjusted based on the user requirements in specific scenarios (such as urban traffic or highway scenarios) to ensure the applicability and reliability of the optimization results.

The reason for the use of fuzzy logic in parameter adjustment is that different application scenarios prioritize performance indicators differently, and these priorities often cannot be precisely expressed with fixed numerical values. Instead, they require adjustment within a fuzzy range. This weight distribution based on fuzzy ranges allows the system to adapt more flexibly and efficiently to various complex communication environments, thereby improving the application effectiveness and reliability of multicarrier modulation techniques in the Internet of Vehicles.

#### 4.4.1. Calculation of Fuzzy Factors

Calculation of the Spectral Efficiency Factor


(23)
$$\eta =\eta_{OFDM-IM}=\left(\log_{2}C_{N}^{K}+K\log_{2}M\right)/\left(N+N_{CP}/G\right).$$


The spectral efficiency here is represented by the spectral efficiency of the OFDM-IM system. When N=K, the system degrades to the classical OFDM system. As η increases, the OFDM-IM system achieves better spectral efficiency, and, when η reaches its maximum, the OFDM-IM system also reaches its maximum spectral efficiency. When the total number of U subcarriers increases, the number of subblocks G = U/N also increases, reducing the CP overhead per subblock and thus improving the spectral efficiency. However, increasing the number of subcarriers per subblock N enriches the subcarrier activation modes but also increases the symbol duration and CP overhead, requiring a trade-off among the system parameters. Furthermore, increasing the length of the cyclic prefix $N_{CP}$ reduces the spectral efficiency, which is the cost of improving the robustness against interference by multiple paths. Since the cyclic prefix is not considered and $N_{CP}$ is set to 0, (Equation 23) simplifies to(24)$$\eta =\eta_{OFDM-IM}=\frac{\log_{2}C_{N}^{K}+K\log_{2}M}{N}.$$

In the IoV, adjusting the spectral efficiency is crucial in meeting various real-time and reliability requirements. While higher spectral efficiency typically improves resource utilization by achieving better bandwidth usage, it may lead to increased interference or energy consumption under certain conditions, such as dense IoV scenarios or energy-constrained devices. Conversely, lower spectral efficiency often results in reduced capacity, potentially affecting the system’s ability to support real-time and critical applications. Therefore, finding the optimal balance of spectral efficiency tailored to specific IoV scenarios is essential. In most cases, traditional OFDM achieves higher spectral efficiency than OFDM-IM because all subcarriers in OFDM are utilized for data transmission, whereas OFDM-IM employs index modulation by activating only a subset of the subcarriers for transmission. This difference gives traditional OFDM an advantage in terms of spectral efficiency but results in significantly higher energy consumption. Moreover, OFDM-IM achieves a significantly lower BER over a wider range of SNRs compared to traditional OFDM. This indicates that high spectral efficiency is not inherently “bad”; rather, it needs to be weighed against the specific application demands. By sacrificing a small amount of spectral efficiency, OFDM-IM demonstrates substantial advantages in terms of energy consumption and the BER. These improvements will be detailed in the subsequent section.

Calculation of the Energy Consumption Factor


(25)
$$NH=NH_{OFDM-IM}=\frac{K}{N}.$$


Energy consumption here is defined as the energy consumption ratio of the OFDM-IM system, represented by the ratio of the number of active subcarriers K to the total number of subcarriers N. Multicarrier index modulation transmits index information by activating only a subset of the subcarriers, thereby reducing the energy consumption. A smaller K/N value indicates lower energy consumption and better system performance.

In the IoV context, proper energy management is crucial. Excessive energy consumption increases power use, drives up the maintenance costs, and may hinder the widespread adoption of IoV technology, complicating the pursuit of low-carbon, eco-friendly travel. Conversely, insufficient energy consumption could limit the performance of IoV devices, affecting the communication quality and stability. For electric vehicles, which rely heavily on batteries, this could impact the driving range and usage time. Therefore, when designing and implementing IoV systems, it is important to strike a balance between energy consumption and performance to promote the healthy development of IoV technology.
Calculation of the BER Factor
(26)$$\;BER=\frac{m_{e}}{m_{t}}=\frac{\sum_{g=1}^{G}m_{e,g}}{m_{t}},$$
where $m_{e}$ represents the number of bits erroneously received in the OFDM-IM index modulation system during transmission. The total number of bits transmitted is $m_{t}$, and the number of bits erroneously received in each subblock is $m_{e,g}$. Therefore, the system’s BER can be calculated using the formula provided above. According to this formula, the smaller the BER factor, the better, as a lower BER signifies higher transmission accuracy, indicating that the system has better energy efficiency.

In the IoV, an appropriate BER is crucial. An excessive BER can lead to packet loss and retransmissions, reducing the transmission speed, increasing the latency, and potentially causing system instability, which in turn raises the energy consumption. Conversely, an overly low BER may result in the waste of system resources through excessive protection and unnecessary data verification, reducing the system’s performance and increasing the costs. Therefore, IoV system design and optimization must carefully consider the impact of the BER to achieve a balance between the BER and communication efficiency, thereby enhancing the overall user experience.

#### 4.4.2. Fuzzification Processing

In fuzzy logic, “fuzzification” is the process of converting the system’s input numerical values into fuzzy values based on pre-established fuzzy membership functions.

As shown in Figure 10, diagram (a) represents the membership function for spectral efficiency in the urban traffic scenario, while diagram (b) illustrates the membership function for spectral efficiency in the highway scenario.

As shown in Figure 11, diagram (a) represents the membership function for energy consumption in the urban traffic scenario, while diagram (b) illustrates the membership function for energy consumption in the highway scenario.

As shown in Figure 12, the membership functions for the BER in the urban traffic scenario are configured identically to those in the highway scenario.

#### 4.4.3. Fuzzy Inference Rules

In the urban traffic scenario, vehicle users often experience good channel quality and convenient charging conditions for electric vehicles. Therefore, when selecting an optimized OFDM scheme, users should comprehensively consider three factors: the spectral efficiency, BER, and energy consumption. In this context, the fuzzy logic system evaluates each factor based on a predefined IF/THEN rule base, as shown in Table 2. The rule base was designed with the thorough consideration of users’ needs for the three factors, particularly the BER, which is especially crucial in urban traffic scenarios. Therefore, when conducting a comprehensive evaluation of multiple factors, a scheme that exhibits optimal performance in terms of the BER should be chosen to ensure reliable and efficient communication.

In highway scenarios, because the number of NLOS components in signal transmission is smaller, the channel quality for vehicle users is generally improved compared to urban traffic scenarios. However, charging challenges for electric vehicles can cause energy consumption anxiety among users, making energy consumption the most important consideration in this scenario. To maximize the energy efficiency, optimization strategies should first ensure low energy consumption and then optimize the bit error rate and spectral efficiency. Therefore, as shown in Table 3, the fuzzy logic rule base prioritizes energy consumption as the primary evaluation criterion during its design. With strong energy consumption performance as a prerequisite, the best index parameter values are then chosen based on the bit error rate and spectral efficiency, ensuring that the communication system is both energy-efficient and maintains the communication quality and speed.

#### 4.4.4. Defuzzification

The fuzzification of output variables is a process that generates numerical results based on the output membership functions and the corresponding degree of point qualification. The definition of the output membership function is shown in Figure 13. Setting the output membership function as a Gaussian curve, compared to linear and trapezoidal curves, helps to smooth the relationship between the input and output variables to a certain extent, thereby reducing the system’s sensitivity.

To obtain an accurate output value, it is necessary to perform precise calculation on the fuzzy output obtained from fuzzy inference. This is achieved through defuzzification using the center of gravity method. The input value is passed through the corresponding membership function to obtain the corresponding output value. The defuzzified value is taken as the x-coordinate of the center of gravity of the shaded area formed by the output variable’s membership function. More specifically, the output membership function can be sliced horizontally according to the corresponding degree, with the top portion removed. Then, the centroid of this shape is calculated, and the x-coordinate of the centroid becomes the defuzzified value. If Y(x) represents the resulting function and *x* denotes the horizontal axis, the center of gravity is calculated as follows:(27)$$\mathrm{COG}=\frac{\int Y(x)xdx}{\int Y(x)dx}.$$

COG represents the suitability of the index parameter in the energy efficiency evaluation system, with higher values indicating better suitability and alignment with the fuzzy logic criteria. In the optimization of OFDM-IM systems based on fuzzy logic, by defining the fuzzy membership functions and rules for the given scenario, the system’s performance for different index parameters can be evaluated. A larger fuzzy logic value indicates that the parameter better meets the energy efficiency maximization requirements of the scenario. By comparing the fuzzy output results for different parameters, the optimal index parameter for the OFDM-IM system in vehicular network scenarios can be determined.

#### 4.4.5. Computational Complexity

The computational complexity of the fuzzy logic method mainly consists of three steps: fuzzification, fuzzy inference, and defuzzification. The complexity of fuzzification is $O(n_{FL})$, where $n_{FL}$ is the number of input variables; the complexity of fuzzy inference is $O(m_{FL})$, where $m_{FL}$ is the number of fuzzy rules; and the complexity of defuzzification is $O(k_{FL})$, where $k_{FL}$ is the number of fuzzy membership functions for the output variable. In this study, the fuzzy logic system includes 3 input variables (BER, spectral efficiency, and energy consumption), 27 fuzzy rules, and 1 output variable. Therefore, the overall complexity is $O\left(n_{FL}+m_{FL}+k_{FL}\right)=O\left(31\right)$. This low complexity makes it well suited to meet the real-time requirements of vehicular communication scenarios.

In comparison, the computational complexity of exhaustive search is $O({C_{ES}}^{d_{ES}})$, where $C_{ES}$ represents the number of candidate parameter values, and $d_{ES}$ is the parameter dimension. While exhaustive search guarantees that the global optimum will be found, its complexity increases exponentially with the growth of the parameter dimensions, making it unsuitable for real-time systems that require rapid decision-making. The genetic algorithm has complexity of $O(g_{GA}\cdot p_{GA})$, where $g_{GA}$ is the number of generations and $p_{GA}$ is the population size. Although it exhibits strong exploratory capabilities in nonlinear optimization, it requires multiple iterations and parameter tuning, resulting in a significant computational overhead. In contrast, the fuzzy logic method, with its parameter adjustment mechanism centered on a rule base, achieves adaptive optimization for dynamic scenarios at a low complexity level of $O(n_{FL}+m_{FL}+k_{FL})$. This makes it particularly suitable for real-time communication scenarios that require rapid decision-making. However, the fuzzy logic method may fall short in terms of global optimization capabilities, as it relies on a predefined rule base, making it challenging to extend to large-scale, high-dimensional parameter spaces. As shown in Table 4, the fuzzy logic method demonstrates significant advantages in terms of applicability, real-time performance, and dynamic scenario optimization, all while maintaining the lowest complexity. On the other hand, exhaustive search and genetic algorithms excel in terms of global optimization capabilities.

## 5. Simulation and Results

This section provides simulation validation for the adaptive regulation of the OFDM-IM model parameters in two representative vehicular network scenarios.

### 5.1. Simulation Results and Analysis for Urban Traffic Scenarios

This section will simulate the user demand- and scenario-adaptive OFDM-IM method in urban traffic scenarios, providing system performance graphs for various values of the index modulation parameters (N, K, M) in these scenarios. The simulation parameters are shown in Table 5. The simulations are conducted following the specifications outlined in the 802.11 p standard [35]. The corresponding subcarrier spacing meets the requirements of vehicular communication scenarios, providing good robustness against Doppler shift and multipath effects.

The fuzzy logic simulation input dataset includes the spectral efficiency, bit error rate, and energy consumption values for different combinations of (N, K, M) under this channel state scenario. The output is the energy efficiency evaluation Y value. The SNR selection is primarily based on the minimum SNR chosen in Section 3. All possible points represent results from arbitrary combinations of different (N,K,M),N≤K, totaling $\left(A_{4}^{4}A_{4}^{4}+A_{8}^{8}A_{4}^{4}+A_{12}^{12}A_{4}^{4}+A_{16}^{16}A_{4}^{4}\right)$ groups. To reduce the computational complexity, combinations with similar results are filtered out, and only one group is analyzed and plotted if it does not significantly affect the overall trend or conclusion. Additionally, for N values of 12 and 16, the discontinuous sampling of the K values can be performed to select the best-performing combinations (N, K, M). Considering the values of the three decisive factors in maximizing the energy efficiency as fuzzy logic inputs, the spectral efficiency, bit error rate, and energy consumption at a 20 dB SNR are entered into the system with predefined membership functions and fuzzy rules. Finally, based on the defuzzified logical output with the maximum value, the condition of energy efficiency maximization is determined, identifying the optimal index parameter point with high spectral efficiency, a low system bit error rate, and low energy consumption. Figure 14 illustrates the system performance for different (N, K, M) values under urban traffic scenarios based on the simulation results.

Figure 14a shows that, under the condition of N = 4 subcarriers, the energy efficiency Y value is maximized at (N, K, M) = (4, 2, 4). Compared to the conventional OFDM point (N, K, M) = (4, 4, 4), the OFDM-IM system at this optimal point achieves 75% of the spectral efficiency of traditional OFDM but has a lower bit error rate and consumes only 50% of the energy, offering superior overall performance. Additionally, compared to the second-best point (N, K, M) = (4, 3, 4), although the spectral efficiency is slightly lower, both the bit error rate and energy consumption improve significantly. The figure also indicates that, as the modulation order M increases, the spectral efficiency gradually improves, but the bit error rate also rises because a higher constellation modulation order M leads to a larger mapping sequence, reducing the minimum Euclidean distance between the symbols on the constellation diagram. Figure 14b illustrates that, with N = 8 subcarriers, the highest energy efficiency Y value is achieved at (N, K, M) = (8, 4, 4). At this optimal point, compared to the conventional OFDM point (N, K, M) = (8, 8, 4), the OFDM-IM system achieves 87.5% of the spectral efficiency of traditional OFDM but with a lower bit error rate and only 50% of the energy consumption, providing better system performance. Compared to the second-best point (N, K, M) = (8, 3, 4), although there are slight deficiencies in the energy consumption and bit error rate, the higher spectral efficiency makes the two points’ energy efficiency values very close, making the second-best point a viable alternative when users are more concerned about energy consumption and the bit error rate. Figure 14c shows that, with N = 12 subcarriers, the highest energy efficiency Y value is achieved at (N, K, M) = (12, 5, 4). Compared to the traditional OFDM point (N, K, M) = (12, 12, 4), this optimal point achieves 79.15% of the spectral efficiency of traditional OFDM but with a lower bit error rate and only 41.6% of the energy consumption, improving the overall system performance. The second-best point (N, K, M) = (12, 9, 2) has overall performance metrics that are inferior to the optimal point. In Figure 14d, with N = 16 subcarriers, the energy efficiency Y value is maximized at (N, K, M) = (16, 6, 4). This optimal point achieves 75% of the spectral efficiency of traditional OFDM (N, K, M) = (16, 16, 4) but with a lower bit error rate and only 37.5% of the energy consumption, delivering better system performance. Compared to the second-best point (N, K, M) = (16, 7, 2), the optimal point has higher spectral efficiency while reducing both the bit error rate and energy consumption, although the second-best point outperforms other points in terms of overall performance, leaving its energy efficiency value second only to the optimal point.

From the simulation results, it can be concluded that the design of the fuzzy logic-based energy efficiency evaluation system is both effective and reasonable for urban traffic scenarios. By analyzing the characteristics and user needs of this scenario and synthesizing the overall performance of the three factors, accurate and reasonable membership functions and fuzzy mapping rules are established. The simulation results ultimately confirm that the optimal index parameters determined by the fuzzy logic output indeed maximize the system performance, allowing the parameters of OFDM-IM to be adaptively regulated according to the scenario.

### 5.2. Simulation Results and Analysis for Highway Scenarios

This section will simulate the OFDM-IM method, based on the user needs and scenario adaptation, for highway scenarios. System performance graphs for different values of the N, K, and M index modulation parameters in highway scenarios will be generated. The simulation parameters are listed in Table 6.

The input dataset for this fuzzy logic simulation includes values of the spectral efficiency, bit error rate, and energy consumption for different (N, K, M) combinations under this channel state scenario. The output is the energy efficiency evaluation Y value. The three decisive factors in maximizing the energy efficiency are used as inputs to the fuzzy logic system. The spectral efficiency, bit error rate, and energy consumption values at an SNR of 15 dB are fed into the system, which has predefined membership functions and fuzzy rules. Based on the defuzzified logical output, the maximum value determines the condition that maximizes the system’s energy efficiency. In this scenario, considering users’ need to manage charging difficulties, the point with the lowest system energy consumption, relatively high spectral efficiency, and a low system bit error rate is identified as the optimal index parameter point. As illustrated in Figure 15, based on the simulation results, the system performance for various index modulation parameter values in highway scenarios is established.

From Figure 15a, it can be seen that, when the number of subcarriers is N = 4, the point with the highest energy efficiency is (N, K, M) = (4, 1, 4). Compared to traditional OFDM, which has (N, K, M) = (4, 4, 4), this optimal point in the OFDM-IM system significantly reduces the bit error rate (BER) and requires only 25% of the energy, even though the spectral efficiency is only 50% of that of traditional OFDM. This configuration sacrifices some spectral efficiency to achieve low energy consumption and a low BER, aligning with the needs of low-energy users and offering superior system performance. Compared to the sub-optimal point (N, K, M) = (4, 2, 2), this optimal point improves both the BER and energy consumption while maintaining the same spectral efficiency, highlighting its comprehensive performance advantage in this scenario. In Figure 15b, where the number of subcarriers is N = 8, the optimal point is (N, K, M) = (8, 2, 4). Although its spectral efficiency is only 50% of that of traditional OFDM (N, K, M) = (8, 8, 4), its BER and energy consumption are significantly lower. Compared to the sub-optimal point (N, K, M) = (8, 3, 4), the optimal point offers better overall performance due to its lower BER and energy consumption, despite its slightly lower spectral efficiency. In the case of N = 12 subcarriers, as shown in Figure 15c, the optimal point (N, K, M) = (12, 4, 4) has spectral efficiency of 66.5% of that of traditional OFDM (N, K, M) = (12, 12, 4). At the same time, it achieves a very low BER and requires only 25% of the energy. This optimal point provides better system performance due to higher spectral efficiency while maintaining low energy consumption and a low BER compared to the sub-optimal point (N, K, M) = (12, 4, 2). Lastly, in the case of N = 16 subcarriers, Figure 15d indicates that the optimal point (N, K, M) = (16, 5, 4) achieves a balance between the spectral efficiency, BER, and energy consumption. At this optimal point, the spectral efficiency of the OFDM-IM system is 75% of that of traditional OFDM (N, K, M) = (16, 16, 4), but its BER is much lower and the energy consumption is only 31.25%. Compared to the sub-optimal point (N, K, M) = (16, 5, 2), the optimal point offers a slightly higher BER but identical energy consumption, while its higher spectral efficiency makes it more suitable for situations where users prioritize spectral efficiency. The sub-optimal point can serve as an alternative option if users place a greater emphasis on minimizing the BER.

The simulation results indicate that the design of the performance evaluation system based on fuzzy logic in highway scenarios is effective, achieved by analyzing the characteristics of the scene and the user requirements. Here, the energy consumption factor is considered the most crucial, and the system balances the BER and spectral efficiency factors while prioritizing energy consumption. The simulation results demonstrate that the optimal index parameters determined through the fuzzy logic output indeed represent the point of maximum system performance, enabling adaptive parameter regulation.

To further verify the BER performance advantages of our proposed fuzzy logic adaptive OFDM-IM parameter optimization method, we conduct an additional set of comparative experiments. Previously, in a fixed urban traffic scenario, we used fuzzy logic to select four optimal parameter combinations: (N, K, M) = (4, 2, 4), (8, 4, 4), (12, 5, 4), (16, 6, 4). To increase the persuasiveness of the results, this supplementary experiment compares these optimal combinations with the typical parameter settings commonly used in traditional OFDM-IM systems.

From [36], we select the widely adopted parameter sets (N, K, M) = (16, 13, 8) and (4, 3, 16) as benchmarks for conventional IM-OFDM. Under identical channel conditions, we plot the BER versus Eb/N0 curves. As shown in Figure 16, compared to the traditional OFDM-IM benchmark parameters, our fuzzy logic-optimized parameter combinations consistently achieve lower BERs across the entire Eb/N0 range. In the medium-to-high Eb/N0 region, our optimal combinations reduce the BER by roughly an order of magnitude relative to the traditional schemes. This indicates that our method not only achieves the comprehensive optimization of the energy efficiency, spectral efficiency, and BER in specific Eb/N0 scenarios, but also demonstrates significant advantages in a more general BER vs. Eb/N0 comparison framework. Through this comparative experiment, we further validate the applicability and superiority of the proposed method in practical vehicular network communication scenarios.

## 6. Conclusions

In this paper, we primarily investigate the challenges associated with applying OFDM-IM technology in vehicular networks, particularly its inability to fully adapt to the complex characteristics of vehicular networks in practical usage and the limitations of the evaluation criteria. Considering the complex and dynamic nature of real-world vehicular networks, we first establish an OFDM-IM transmission model based on a specific vehicular network scenario and formulate an energy efficiency maximization equation. The energy efficiency is influenced by the spectral efficiency, BER, and energy consumption. We propose a vehicular network OFDM-IM communication optimization method based on user requirements and scenario adaptability. This approach flexibly allocates the weights of the three factors through fuzzy logic according to different user requirements and scenarios. The simulation results demonstrate that this method enables the selection of the optimal parameters in various vehicular network scenarios according to different user requirements, thus optimizing the communication performance.

## Figures and Tables

**Figure 1 sensors-25-00663-f001:**
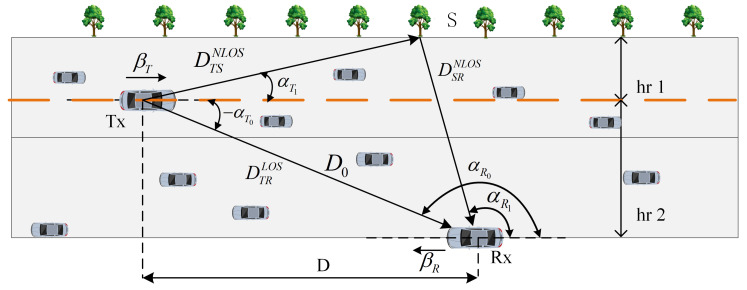
V2V communication in a typical straight road scenario, including scatterers on both sides of the road, as well as LOS and NLOS components.

**Figure 2 sensors-25-00663-f002:**
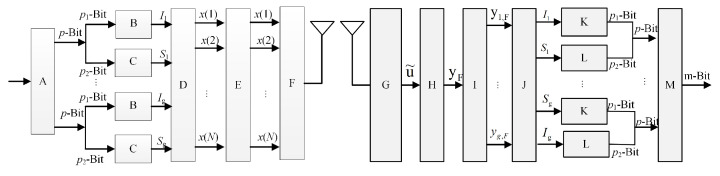
Block diagram of transmitter and receiver modules.

**Figure 3 sensors-25-00663-f003:**
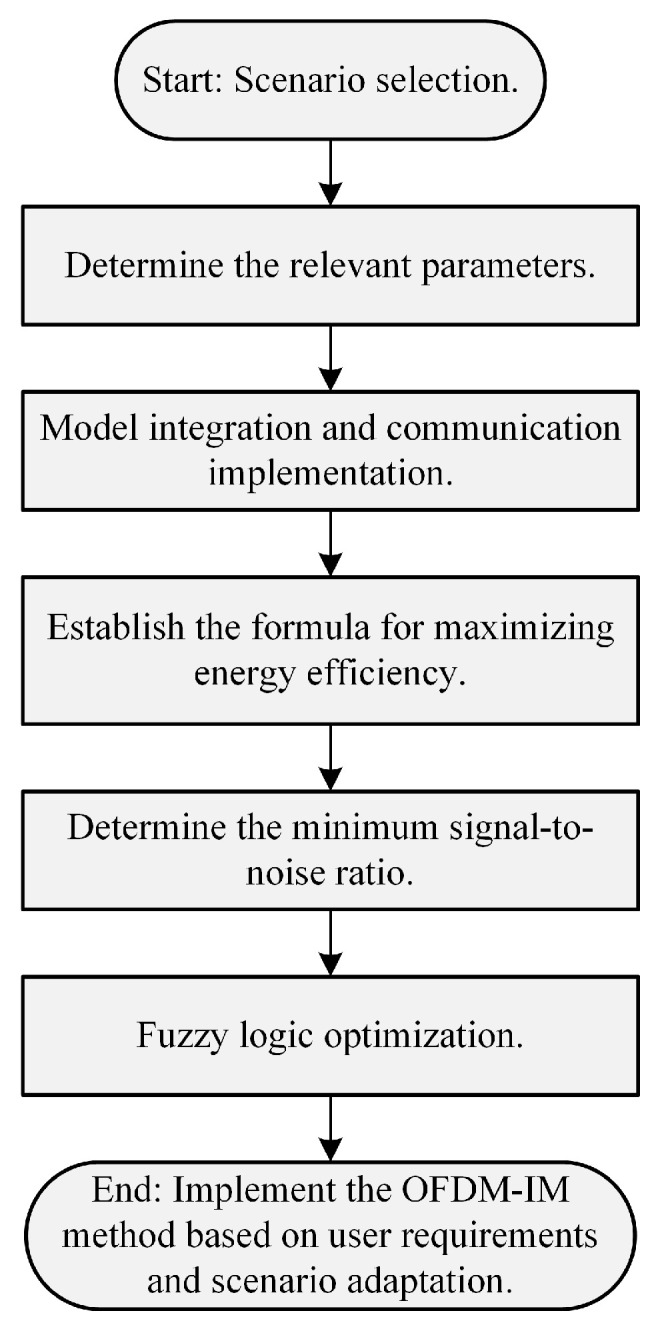
Block diagram of OFDM-IM method that is adaptive to user requirements and scenario.

**Figure 4 sensors-25-00663-f004:**
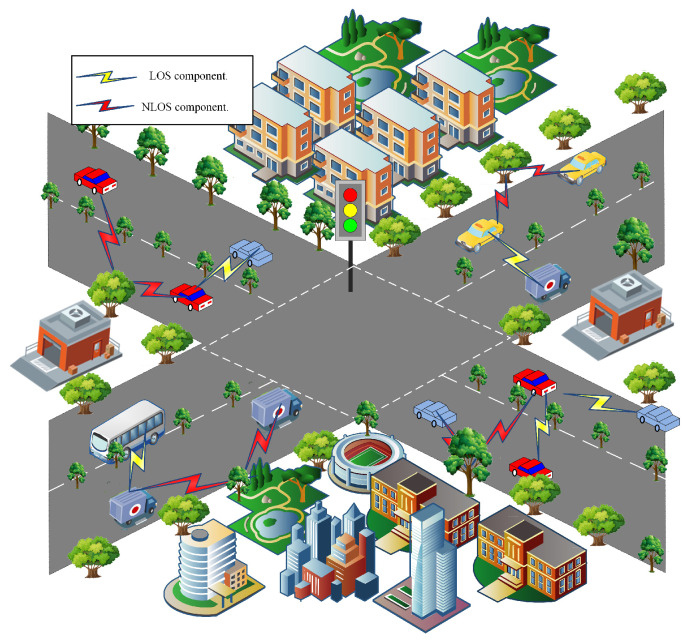
OFDM-IM communication in highway scenarios.

**Figure 5 sensors-25-00663-f005:**
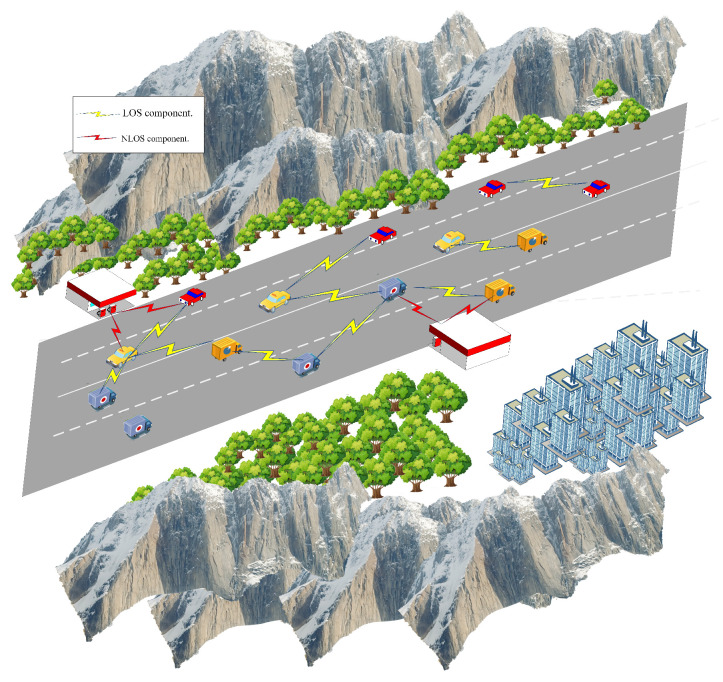
OFDM-IM communication in urban traffic scenarios.

**Figure 6 sensors-25-00663-f006:**
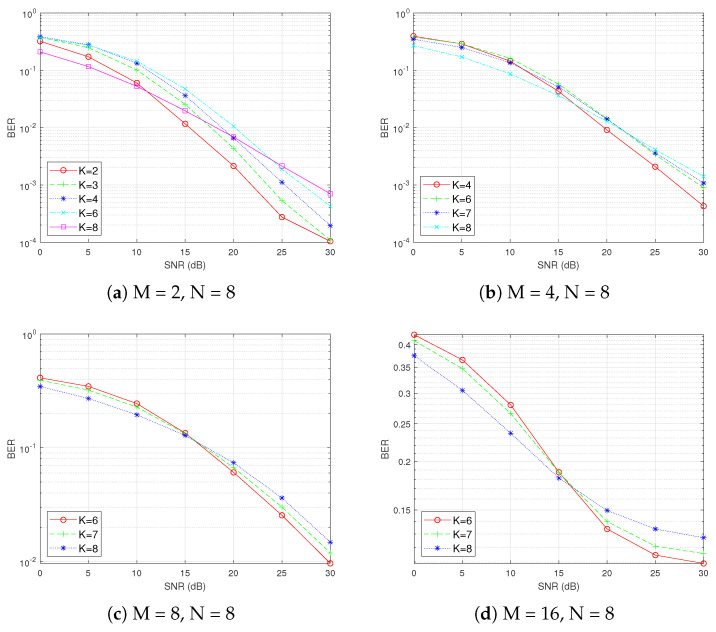
Fixing N and varying M and K.

**Figure 7 sensors-25-00663-f007:**
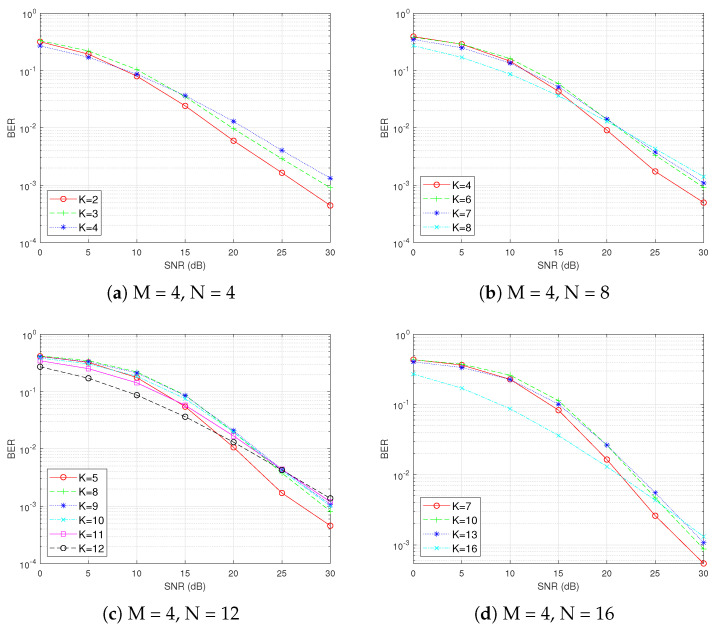
Fixing M and varying N and K.

**Figure 8 sensors-25-00663-f008:**
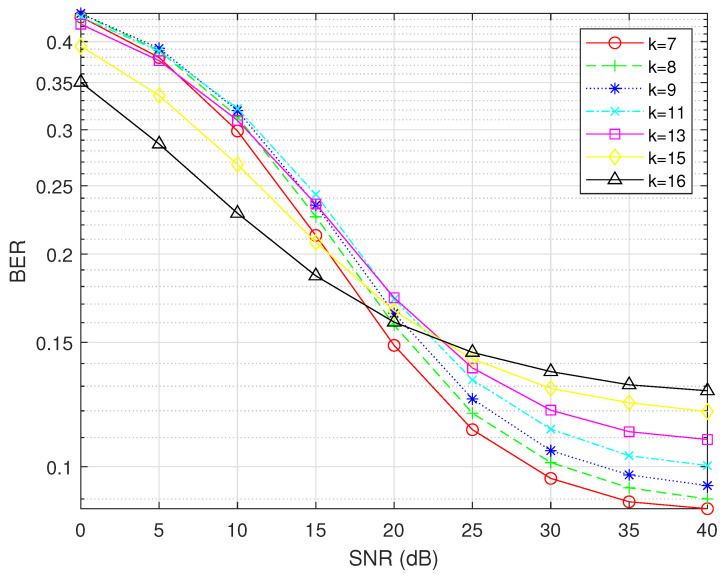
Trend of the BER versus SNR in the urban traffic scenario.

**Figure 9 sensors-25-00663-f009:**
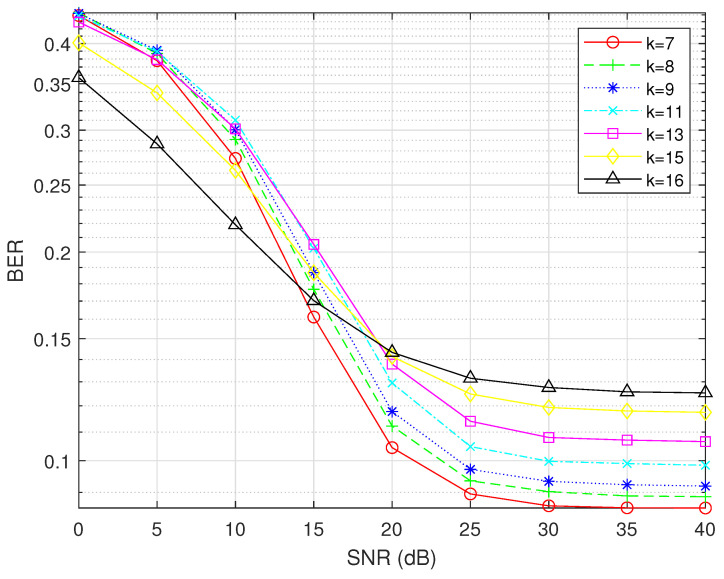
Trend of the BER versus SNR in the highway scenario.

**Figure 10 sensors-25-00663-f010:**
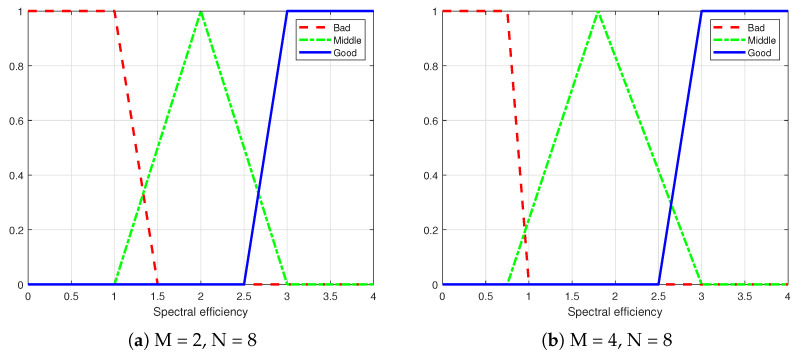
Membership functions for spectral efficiency in different scenarios.

**Figure 11 sensors-25-00663-f011:**
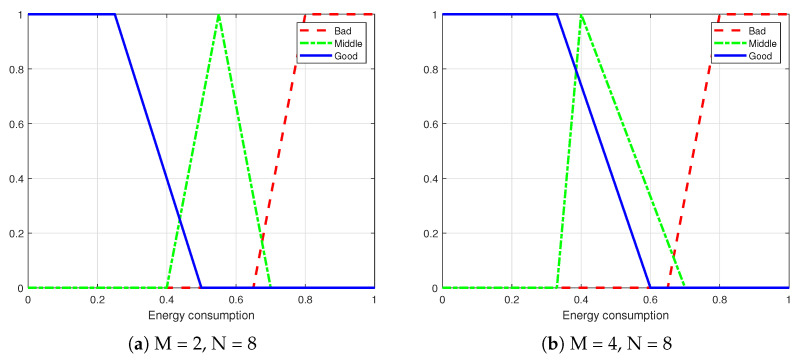
Membership functions for energy consumption in different scenarios.

**Figure 12 sensors-25-00663-f012:**
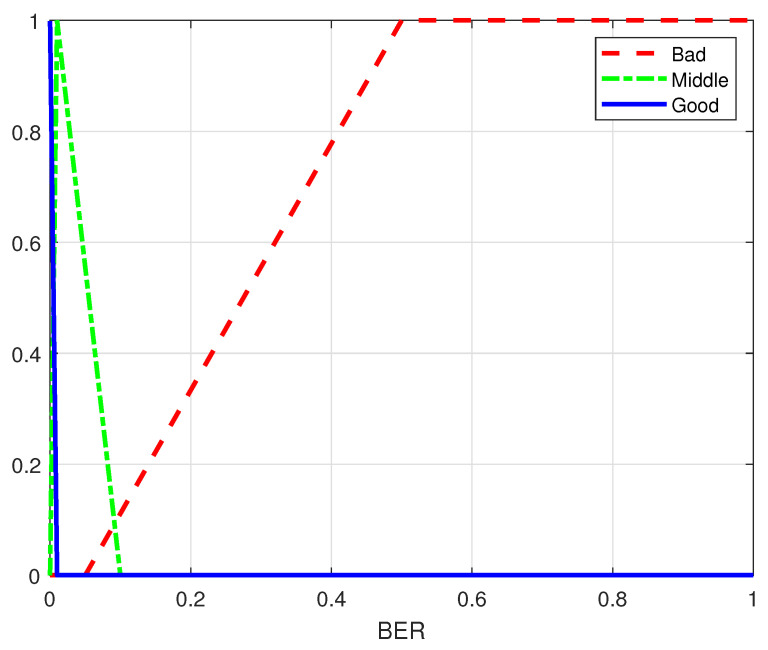
Membership functions for BER in different scenarios.

**Figure 13 sensors-25-00663-f013:**
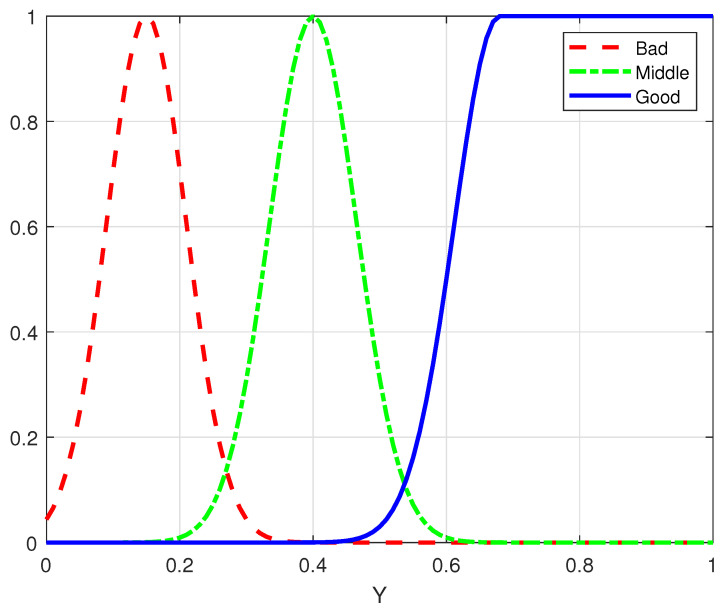
Membership function of fuzzy system output.

**Figure 14 sensors-25-00663-f014:**
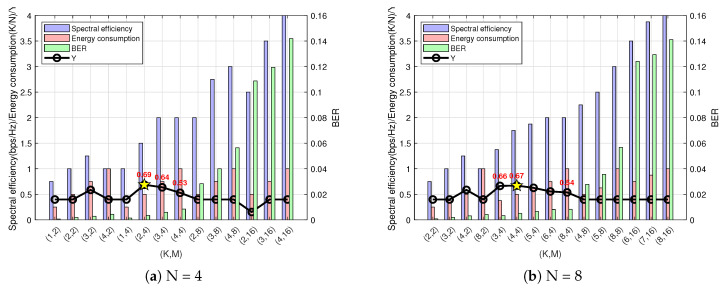
Output results chart for urban traffic scenario. The star indicates the maximum point of the Y value.

**Figure 15 sensors-25-00663-f015:**
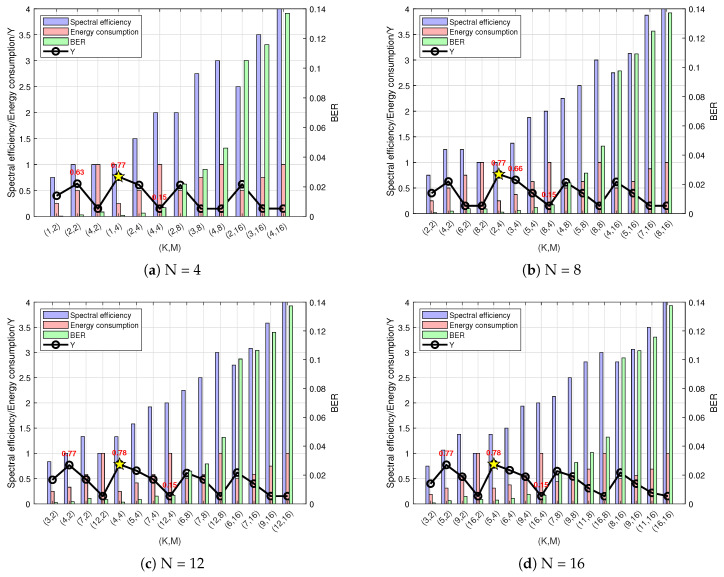
Output results chart for highway scenario. The star indicates the maximum point of the Y value.

**Figure 16 sensors-25-00663-f016:**
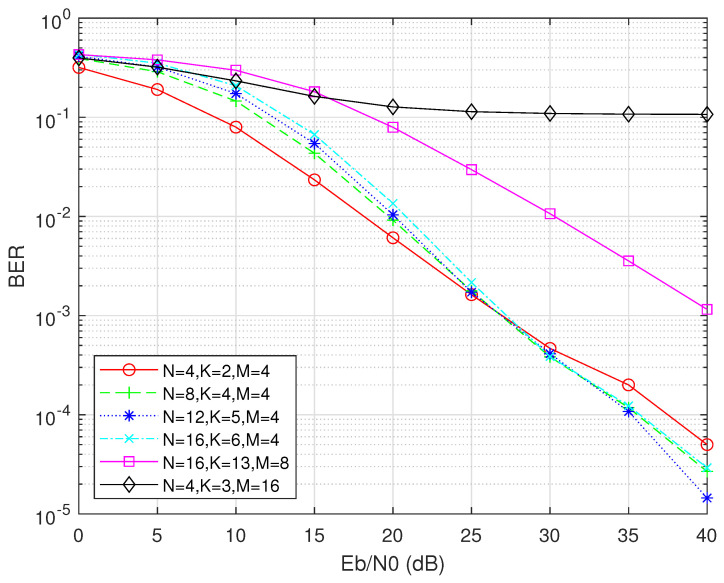
Performance comparison of BER versus Eb/N0 between adaptive OFDM-IM and conventional OFDM-IM systems.

**Table 1 sensors-25-00663-t001:** Channel model parameters.

Model Parameter	Definition
hr1	The distance between $T_{X}$ and S.
hr2	The distance between $R_{X}$ and $T_{X}$.
D	The horizontal straight-line distance between $T_{X}$ and $R_{X}$.
$D_{0}$	The straight-line distance between $T_{X}$ and $R_{X}$.
$V_{T_{x}}/V_{R_{x}}$	The speed of the signal transmitting/receiving vehicle.
$\beta_{T}/\beta_{R}$	The direction of travel of the signal-transmitting/receiving vehicle.
$\alpha_{T_{0}}/\alpha_{R_{0}}$	The angle of the direct component for transmission/reception.
$\alpha_{T_{1}}/\alpha_{R_{1}}$	The signal’s angle of departure/angle of arrival.
$D_{TR}^{LOS}$	The line-of-sight propagation path between $T_{X}$ and $R_{X}$.
$D_{TS}^{NLOS}$	The propagation path between $T_{X}$ and S.
$D_{SR}^{NLOS}$	The propagation path between S and $R_{X}$ forms a single-reflection non-line-of-sight propagation path with $D_{TS}^{NLOS}$.

**Table 2 sensors-25-00663-t002:** IF/THEN rule base for urban traffic scenario.

Rule	Spectral Efficiency Factor	BER Factor	Energy Consumption Factor	Level
1	Good	Good	Good	Good
2	Good	Good	Moderate	Good
3	Good	Good	Bad	Moderate
4	Moderate	Good	Good	Good
5	Bad	Good	Good	Moderate
6	Good	Moderate	Good	Good
7	Good	Bad	Good	Moderate
8	Good	Moderate	Moderate	Good
9	Good	Moderate	Bad	Moderate
10	Good	Bad	Moderate	Moderate
11	Good	Bad	Bad	Moderate
12	Moderate	Good	Moderate	Good
13	Moderate	Good	Bad	Good
14	Bad	Good	Moderate	Good
15	Bad	Good	Bad	Moderate
16	Moderate	Moderate	Good	Moderate
17	Moderate	Bad	Good	Moderate
18	Bad	Moderate	Good	Moderate
19	Bad	Bad	Good	Moderate
20	Moderate	Moderate	Moderate	Moderate
21	Moderate	Moderate	Bad	Moderate
22	Moderate	Bad	Moderate	Bad
23	Bad	Moderate	Moderate	Moderate
24	Moderate	Bad	Bad	Bad
25	Bad	Moderate	Bad	Moderate
26	Bad	Bad	Moderate	Bad
27	Bad	Bad	Bad	Bad

**Table 3 sensors-25-00663-t003:** IF/THEN rule base for highway scenario.

Rule	Spectral Efficiency Factor	BER Factor	Energy Consumption Factor	Level
1	Good	Good	Good	Good
2	Good	Good	Moderate	Moderate
3	Good	Good	Bad	Bad
4	Moderate	Good	Good	Good
5	Bad	Good	Good	Moderate
6	Good	Moderate	Good	Good
7	Good	Bad	Good	Moderate
8	Good	Moderate	Moderate	Good
9	Good	Moderate	Bad	Bad
10	Good	Bad	Moderate	Moderate
11	Good	Bad	Bad	Bad
12	Moderate	Good	Moderate	Moderate
13	Moderate	Good	Bad	Bad
14	Bad	Good	Moderate	Moderate
15	Bad	Good	Bad	Bad
16	Moderate	Moderate	Good	Good
17	Moderate	Bad	Good	Good
18	Bad	Moderate	Good	Good
19	Bad	Bad	Good	Moderate
20	Moderate	Moderate	Moderate	Moderate
21	Moderate	Moderate	Bad	Bad
22	Moderate	Bad	Moderate	Moderate
23	Bad	Moderate	Moderate	Moderate
24	Moderate	Bad	Bad	Bad
25	Bad	Moderate	Bad	Bad
26	Bad	Bad	Moderate	Moderate
27	Bad	Bad	Bad	Bad

**Table 4 sensors-25-00663-t004:** Comparison of complexity, advantages, and disadvantages of three algorithms.

Method	Complexity	Advantages	Disadvantages
Fuzzy Logic Method	$O(n_{FL}+m_{FL}+k_{FL})$	Efficient, suitable for real-time scenarios, strong adaptability in dynamic scenarios	Limited adaptability in global optimization
Exhaustive Search	$O({C_{ES}}^{d_{ES}})$	Guarantees global optimal solutions	High computational complexity for high-dimensional problems, unsuitable for real-time systems
Genetic Algorithm	$O(g_{GA}\cdot p_{GA})$	Balances exploration and efficiency, suitable for nonlinear optimization problems	Requires multiple iterations, relatively high computational overhead

**Table 5 sensors-25-00663-t005:** Simulation parameters for urban traffic scenarios.

Simulation Parameter	Configured Value
Rician Factor R	10 dB
Reflection Coefficient μ	0.5
Number of Subcarriers per Subblock N	4/8/12/16
Number of Active Subcarriers K	1-N
Modulation Order of Constellation M	2/4/8/16
SNR Value	20 dB
Total OFDM Subcarriers U	64
Number of Subblocks G	U/N
Bandwidth	10 MHz
Length of the CP $N_{\mathrm{CP}}$	16

**Table 6 sensors-25-00663-t006:** Simulation parameters for highway scenarios.

Simulation Parameter	Configured Value
Rician Factor R	20 dB
Reflection Coefficient μ	0.2
Number of Subcarriers N	4/8/12/16
Number of Active Subcarriers K	1-N
Modulation Order of Constellation M	2/4/8/16
SNR Value	15 dB
Total OFDM Subcarriers U	64
Number of Subblocks G	U/N
Bandwidth	10 MHz
Length of the CP $N_{\mathrm{CP}}$	16

## Data Availability

The raw data supporting the conclusions of this article will be made available by the authors on request.

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
