# Peer review of "Traffic and Scenario Adaptive OFDM-IM for Vehicular Networks: A Fuzzy Logic Based Optimization Approach"

_sensors, 2025, doi:10.3390/s25030663_

Round 1

Reviewer 1 Report

Comments and Suggestions for Authors

The paper addressed a relatively emergent technology of OFDM-IM for V2V communications, focussing on two aspects

1) provide a simplified straight encounter V2V channel model 

2) Provide a methodology for V2V OFDM-IM parameter optimization with multi-metric approach (BER, SNR, spectral efficiency)

The paper is well written, the methodology is well explained and results are mostly clear.

My main concerns with the content of the manuscript is in the system model and the simulation parameters as follows:

1) Channel model in (1) - (10) depends only on t variable implying that only narrowband fading is being considered?, however,  the system model in (16) with H being a diagonal matrix accounts for a Wideband (frequency selective) channel model in which ICI is not present. This scenario is inconsistent with the main purpose of OFDM-IM technology. 

The simulation parameters employ a very low subcarrier number of 16 (maximum) this implies a very high subcarrier spacing, the authors must provide information of the assumed channel coherence bandwidth in the model that justifies this parameters selection. Also elaborate on the spectral efficiency penalization of such N considering other variables like the CP length and pilot subcarriers.

Minor observation: page two line 65 reads "We proposes"

Reviewer 2 Report

Comments and Suggestions for Authors

The work proposes an adaptive method to optimize OFDM-IM in vehicular networks, addressing spectral efficiency, bit error rate, and energy consumption. A vehicular communication channel is modeled, and parameters are adjusted according to user requirements and scenarios using fuzzy logic.

Please consider the following comments to improve the work:

Section 5. Simulation and Results:

  • Include all configuration parameters of the evaluated system, such as the FFT block size and CP, in accordance with the frame defined in the 802.11p standard, as described in:
    Del Puerto-Flores, J.A.; Castillo-Soria, F.R.; Vázquez-Castillo, J.; Palacio Cinco, R.R. Maximal Ratio Combining Detection in OFDM Systems with Virtual Carriers Over V2V Channels. Sensors 2023, 23, 6728. https://doi.org/10.3390/s23156728.
  • Compare the proposed system's performance against conventional IM-OFDM, particularly in BER vs Eb/N0 tests.
  • Perform an analysis of the computational complexity (Big O) of the proposed algorithms compared to existing systems in the state of the art. This analysis would provide a more comprehensive view of performance, not only in terms of throughput and delay but also computational efficiency, which would benefit both the reviewers and the readers.

Conclusions:

  • Include specific data and simulation results to support the claims made. This would help provide a solid foundation for evaluating the impact of the proposed algorithm and better contextualize its benefits compared to other systems.

Reviewer 3 Report

Comments and Suggestions for Authors

The paper proposes a user-demand and scenario-adaptive OFDM-IM method for V2X communications, addressing limitations of existing systems by optimizing index parameters based on spectral efficiency, BER, and energy consumption, with a fuzzy logic-based energy efficiency framework and a V2V channel model for vehicular environments, demonstrating improved system reliability, efficiency, and flexibility through simulation results. However, there are some issues to be addressed.

1.      Some statements in the manuscript are not entirely precise. For instance, the authors mention that OFDM-IM technology was proposed in recent years, yet most of the cited references date back nearly a decade.

2.      The authors state that “OFDM-IM can improve signal transmission efficiency and fully utilize spectral resources.” However, in reality, compared to OFDM systems that utilize the full set of subcarriers, the spectral resource utilization of OFDM-IM is actually less efficient.

3.      The distinction between the IM-based MIMO-OFDM system in Reference [6] and the traditional OFDM-IM system is not clearly reflected in the introduction.

4.      The authors’ statements lack logical consistency. For instance, the claim that “Existing OFDM-IM systems primarily use low bit error rate as the evaluation criterion, neglecting other key performance indicators such as spectral efficiency and energy consumption” contradicts the assertion that “OFDM-IM can improve signal transmission efficiency.” Similarly, the statement that “The current OFDM-IM systems evaluate performance solely based on low BER, neglecting important factors such as spectral efficiency and power consumption [20–23]” appears inconsistent with the subsequent claim that “Multidimensional index modulation, which utilizes both spatial and frequency domains for index modulation, was extensively studied, significantly enhancing system transmission capacity.”

5.      The manuscript contains grammatical and punctuation issues, such as the absence of periods after many equations. Please carefully review and correct these errors.

6.      None of the figures in the manuscript are vector graphics. It is recommended to use vector graphics to ensure clarity and scalability.

7.      The description of the system model in Section 3.3 is not sufficiently clear and can be quite confusing. It is recommended to provide a more detailed and structured explanation to enhance readability and understanding.

8.      Some abbreviations, such as LLR, are not explained. It is recommended to provide full definitions for all abbreviations when they first appear to ensure clarity for readers.

9.      In Equation (18), the metrics have different dimensions. Should normalization be considered to ensure consistency and comparability?

10.   The authors conclude from Figure 6 that “the minimum SNR required for the BER of OFDM-IM to surpass traditional OFDM gradually increases.” However, the figure does not clearly support this conclusion, and it is unclear whether the performance depicted corresponds to the OFDM or OFDM-IM system. This ambiguity should be addressed to improve the validity of the analysis.

11.   The manuscript states that “an SNR of 20 dB is chosen.” However, this seems questionable and conflicts with the statement that “SNR may be limited due to multipath fading, reflections, and other types of interference”. It is recommended to provide a justification for this choice or discuss its feasibility in practical scenarios.

12.   The authors state that “In the IoV, reasonable adjustments to spectral efficiency are crucial for meeting the real-time and reliability needs of various applications. Excessively high spectral efficiency could lead to wasted resources, while too low a value might affect the real-time performance of critical applications such as autonomous driving. Thus, it is essential to find an optimal balance point for spectral efficiency according to specific requirements, ensuring that IoV communication is both efficient and stable.” However, this analysis appears inconsistent with practical realities. High spectral efficiency typically improves resource utilization rather than causing waste, while low spectral efficiency often leads to insufficient capacity rather than solely affecting real-time performance. A more accurate and nuanced discussion of the trade-offs in spectral efficiency for IoV applications is recommended.

13.   The manuscript lacks an explanation of the principles and methodology of fuzzy logic. The definition of membership functions is unclear, and the approach used in Figures 10–12 is not described. It is recommended to include a detailed explanation of the fuzzy logic framework and clarify how the membership functions were defined and applied in the figures.

14.   The manuscript lacks an analysis of the computational complexity of the fuzzy logic optimization algorithm. It is recommended to include a detailed discussion on the algorithm’s complexity to assess its feasibility for practical implementation in vehicular communication systems.

Comments on the Quality of English Language

The english expression should be improved.

Round 2

Reviewer 1 Report

Comments and Suggestions for Authors

The authors have addressed most of this reviewer comments in the current version of the manuscript, this reviewer have no further concerns.

Reviewer 2 Report

Comments and Suggestions for Authors

It would be beneficial to include a reference supporting the values taken for the conventional OFDM system used in the study. This would not only provide greater clarity and traceability of the configured parameters but also strengthen the technical foundation of the research. A suitable reference is:

Del Puerto-Flores, J.A.; Castillo-Soria, F.R.; Vázquez-Castillo, J.; Palacio-Cinco, R.R. Maximal Ratio Combining Detection in OFDM Systems with Virtual Carriers Over V2V Channels. Sensors 2023, 23, 6728. https://doi.org/10.3390/s23156728

Incorporating this citation will provide readers with a stronger context on the foundations of the evaluated system and facilitate the replication or comparison of results in future studies.

Reviewer 3 Report

Comments and Suggestions for Authors

The paper proposes a user-demand and scenario-adaptive OFDM-IM method for V2X communications, addressing limitations of existing systems by optimizing index parameters based on spectral efficiency, BER, and energy consumption, with a fuzzy logic-based energy efficiency framework and a V2V channel model for vehicular environments, demonstrating improved system reliability, efficiency, and flexibility through simulation results. The authors have made revisions in response to the reviewer’s comments, but some issues remain in the manuscript.

1. Some minor inconsistencies in punctuation were observed. For instance, a period should follow Equation 1. Additionally, spaces are missing after some punctuation marks. A careful review of the punctuation throughout the manuscript would be beneficial.

2. The manuscript contains some grammatical errors. For instance, the parameter descriptions after Equation 2 should be preceded by 'where'.

3. Equations 3 and 4 appear to be redundant.

4. The caption for Figure 2 is too long. The explanation of the module could be incorporated into the main text.

5. Abbreviations should be defined upon their first appearance in the text, e.g., OFDM-IM.

6. Consider revising the structure of Section 4.4. It would be more effective to explain the fundamentals of fuzzy logic before presenting the reasons for its adoption.
